# Comparing the Effects of Class Origins versus Race in the Intergenerational Transmission of Poverty

**Arthur Sakamoto** [1,*] , **Li Hsu** [2] **and Mary E. Jalufka** [1]

1    Department of Sociology, Texas A&M University, College Station, TX 77843, USA; maryjalufka@gmail.com
2    Department of Sociology, University of Wisconsin, Madison, WI 53706, USA; lhsu22@wisc.edu
*    Correspondence: asakamoto@tamu.edu

**Abstract:** Building upon prior research on intergenerational income mobility, we assess class effects versus racial effects on the probability of becoming a poor adult, broken down by gender. We define the class effect (for each race-and-gender group) as the difference between the probability that a person who was born into the lowest income quintile becomes poor and the probability that a person who was born into the highest income quintile becomes poor. For each minority-by-gender group, using Whites as the baseline, the racial effect is defined as the average racial differential in the probability of becoming a poor adult, irrespective of class origins. The results indicate that, for all minority-by-gender groups, the class effect is larger than the racial effect. Our findings underscore the continuing significance of the comparatively large effects of class origins, which have not been adequately acknowledged in recent research.

**Keywords:** poverty; racial inequality; class inequality; intergenerational income mobility

## 1. Social Stratification in America: The Enduring Significance of Race and Poverty

Poverty continues to persist in the United States, despite the general trend towards continued economic growth (Brady and Jäntti 2016). This pattern is evident even with regard to the US Census's official poverty measure, which is an absolute measure of poverty that depends only on the absolute (i.e., not relative) level of income (Iceland 2006). Because the benefits of economic growth tend to disproportionately accrue to the upper two quintiles of the income distribution, the official poverty rate has been relatively stagnant while household income inequality has been increasing in recent decades (Brady and Jäntti 2016; Economic Policy Institute 2021; Iceland 2006). In 2019, the Gini coefficient for household income inequality in the United States was 0.484 (Semega et al. 2020), which is higher than other developed nations (Thevenot 2017).

In addition to the persistence of poverty and a higher level of economic inequality, another endemic feature of social stratification is the income differentials by racial groups in American society (Iceland 2017). Much of the focus has been on African Americans, who have most notably faced a long history of discrimination (Wilson 1980). Other major racial categories that are often considered are Native Americans, Hispanics, and Asian Americans (Marger 2015). Of these groups, Native Americans have the longest history, while most of the growth of the Hispanic and Asian American populations has occurred in recent decades (Iceland 2017). African Americans, Native Americans, and Hispanics all have lower average socioeconomic outcomes, while Asian Americans tend to be on par with or even higher than Whites (Iceland 2017; Marger 2015; Sakamoto et al. 2009).

Because both poverty and racial minority status are notable disadvantages (with the exception of Asian Americans with regard to minority differentials), our analysis seeks to clarify which disadvantage has a greater overall total negative effect on the chances for upward intergenerational income mobility. That is, we seek to clarify whether childhood poverty or racial minority status, per se, is more consequential in reducing

one's "life chances" (Wilson 1980, p. 1), or, in other words, a favorable income position in contemporary American society. Most persons may be characterized as having been born poor (or not) and as having been born a racial minority (or not). Our objective is to assess whether having had an impoverished childhood is more or less than the disadvantage that is associated with racial minority status.

Poverty status and racial minority status are correlated because African Americans, Native Americans, and Hispanics, especially, have significantly higher poverty rates than Whites (Iceland 2017). Our analysis seeks to assess the net effects of poverty status and minority status, per se, by calculating the disadvantage of poverty for each minority group separately, and then comparing that with the disadvantage of minority status overall. In other words, for each minority group considered separately, is their general disadvantage vis-à-vis Whites larger than their disadvantage of being born poor vis-à-vis wealthy (among members of the same minority group)? For each minority group, the differential "life chances" associated with being born poor versus being born wealthy may be considered as a particular type of class effect because the differential is calculated within the given minority group so that it is inherently "controlling for" race.

## 2. The Resurgence of Critical Race Theory in American Sociology

Our research interest is in comparing the minority effect with the class effect (i.e., in terms of being born poor) for each racial-by-gender group because this important issue has not been adequately investigated in recent research. For example, Baker et al. (2021) emphasize "the enduring significance" of racial effects on poverty and pay little attention to the issue of class effects. While we agree that racial effects are generally significant, our investigation provides further insight by placing them into a broader context by comparing their sizes relative to the effects of class origins, rather than simply ignoring the latter (e.g., Bonilla-Silva 2004). Sociological theory and research should not be restricted to considering just one variable, such as race. A more informative analysis should go beyond ascertaining whether a certain net effect is greater than zero and also investigate which nonzero net effects are greater than other nonzero net effects.

In general, critical race theories have proliferated and have become highly influential in recent years in American sociology (e.g., Alexander 2012; Baker et al. 2021; Bonilla-Silva 2004, 2015; Bonilla-Silva and Baiocchi 2001; Bonilla-Silva and Dietrich 2011; Christian et al. 2019, 2021; Conley 2010; Feagin 2013, 2020; Ray et al. 2017). The common theme in these studies is the assumption that minority disadvantages are greater than class disadvantages. The primary focus is on minorities as being more likely to be lower income. Middle-class and wealthy minorities are only considered in terms of also having faced racism (e.g., Feagin 1991; Ladson-Billings 2009). Class effects among minority groups themselves are ignored, and are, thus, implicitly assumed to be small and substantively insignificant. Our analysis of intergenerational mobility fills an important lacuna because our study is the first to ascertain class effects among minority groups (in terms of the advantages of the affluent versus the poor), and to investigate the size of these net class effects relative to overall minority disadvantages.

## 3. Data and Methods

Class effects are various because different points in the distribution of income may be compared. In the following, we focus on the contrast between lower-income households versus higher-income households. The former is commonly referred to as poor, while the latter is often considered to be affluent. In focusing on this contrast, we do not mean to imply that other class contrasts are unimportant, but only that they may be further explored in future research. We investigate the contrast between poverty versus affluence because it is the clearest manifestation of the rising class inequality in the United States, as was discussed above.

The main source of data for our analysis are the reported findings of Chetty et al. (2020), including their Online Appendix. Chetty et al. (2020) is a monumental, path-

breaking study that has been cited over 800 times in only two years.[1] Chetty et al. (2020) is based on unprecedented access to literally hundreds of millions of records of confidential administrative data. The inordinately large quantity of data is needed in order to accurately ascertain long-term income, and especially for minorities, whose sample sizes are generally too small to be adequately represented in the available longitudinal datasets. Teams of researchers and computer programmers were needed to complete this research, and a great deal of funding for it was provided by a variety of agencies and foundations (Chetty et al. 2020, p. 203). While some prior research has investigated the available longitudinal data with much smaller samples to study White versus Black differentials (Bhattacharya and Mazumder 2011; Bloome 2014), Chetty et al. (2020) is the first (and, thus far, only) research that uses data for millions of Americans that provides detailed intergenerational income mobility statistics for other minority groups as well. Given its very high quality and notably important substantive significance, further refined consideration of Chetty et al.'s (2020) statistical findings is merited.

The three main data sources utilized by Chetty et al. (2020) include the 2000 and the 2010 US Censuses, the 2005–2015 American Community Surveys, and federal income tax records, which are maintained by the US Internal Revenue Service. Persons were first identified by their Social Security numbers, and their records were then anonymously linked across these data sources. After incorporating measures for parents' and their children's incomes, the final sample size investigated by Chetty et al. (2020) amounts to over twenty million Americans, including both citizens and US permanent residents.

The target population is all persons who were born in the United States between 1978 and 1983, and who were claimed as a child dependent on a US federal income tax form in at least one year from 1994 to 2015. To be included in the sample, the parents needed to be residing in the United States legally, while the offspring also needed to be linked to their parents' tax data from 1994, 1995, and 1998 through 2000. Finally, the offspring's own income needed to be linked to W-2 forms for 2014 and 2015, when this cohort was between the ages of 31 and 37 years.

The parental income was defined as the gross total household income (that is, pre-tax earnings), according to their federal tax filings for the years 1994, 1995, and 1998 through 2000. The offspring income was similarly defined in terms of pre-tax earnings, although, as noted above, this information was sourced from W-2 forms rather than federal income tax forms. To estimate the income mobility, the earnings of the parents and children were ranked relative to those in the same cohort group (that is, other parents and children during the collection periods). In doing so, the comparative mobility between children and parents is always framed in reference to the earnings of their peers during the same timeframe.

With regard to our operationalization of poverty, the limitations of the US Census official measure have been documented, and important research has been conducted on the conceptualization and measurement of different definitions of poverty (Brady 2003). We do not mean to gainsay the significance of this prior research, but, for our research purposes, in the following, we generically use the term "poverty" to refer to the bottom quintile of the distribution of long-term household income. Defined as having long-term income in the bottom quintile, we can ascertain poverty rates for different racial groups as directly reported by Chetty et al. (2020), including in the Online Appendix.

Poverty in contemporary developed nations has a major relative component (Brady 2003; Rainwater and Smeeding 2003; Smeeding 2016). Abundant evidence indicates that households whose money income is appreciably below the median have long been socially viewed as poor people in American society (e.g., Isenberg 2017; Lynd and Lynd 1937; Pals and Kaplan 2013; Rainwater 1974; Warner and Lunt 1941). In the contemporary United States, households in the bottom quintile of the distribution of income would typically be considered poor by many statistical measures of relative poverty (Iceland 2006; Smeeding 2016). Since the US Census official poverty hovered around from 12% to 15% for the time period of Chetty et al.'s data (1989 through 2015), many households in the bottom quintile would likely be considered poor by the official definition as well.[2]

While we refer to the lowest quintile of long-term household income as poverty, we use the term "affluence" to refer to the highest quintile of long-term household income. As mentioned above, we focus on class in terms of the contrast between the poor versus the affluent. Operationalizing these class positions in terms of long-term income quintiles, we define the class effect as the difference between the probability that a person who was born into the lowest income quintile becomes a poor adult and the probability that a person who was born into the highest income quintile becomes a poor adult. This calculation refers to the advantage (i.e., lower probability) that a child born into affluence has over a child born into poverty in terms of becoming a poor adult. This class effect is calculated separately for each minority group so that race is "held constant." In other words, for each racial group separately considered, we estimate the higher probability that a person born into a poor household has on the probability of becoming a poor adult compared to a person born into an affluent household (i.e., where poverty and affluence are based on the quintile distribution of long-term incomes, as described above).

Defined in this way, the class effect may be calculated using the published findings in Chetty et al. (2020). We use their reported statistics, which are further described below. By utilizing their detailed results, which are provided in the Online Appendix of Chetty et al. (2020), we are able to furthermore break down the class effects by both gender (i.e., male versus female) and racial category simultaneously (i.e., non-Hispanic Whites, Hispanics, non-Hispanic African Americans, non-Hispanic Asian Americans, and non-Hispanic Native Americans). Hereafter, we refer to these racial categories as Whites, Hispanics, Blacks, Asians, and Native Americans, respectively.

For each minority group, a total racial effect is defined as the average racial differential in the probability of becoming a poor adult, irrespective of class origins. Using Whites as the baseline category, the racial effect is the difference in the probability that a person who was born into the $j$th minority group becomes a poor adult and the probability that a person who was born into the White group becomes a poor adult. This calculation thus refers to the overall racial differential between a White child and the $j$th minority child in the chances of becoming a poor adult.

In summary, the probability of adult poverty if born into poverty is a percentage, as reported by Chetty et al. (2020), and may be denoted as Pa | Po. We designate this quantity as Pa | Po, meaning poverty (P) as an adult (a) given poverty (P) in origins (o). The probability of affluence if born into poverty is another percentage that is also reported by Chetty et al. (2020) and may be denoted as Pa | Ao. We designate this quantity as Pa | Ao, meaning poverty (P) as an adult (a) given affluence (A) in origins (o). The class effect is then equal to (Pa | Po–Pa | Ao), which refers to the difference in the percentage (i.e., probability) of poor kids versus affluent kids becoming poor adults. This difference is calculated separately for each racial group so that minority status is not a confounding factor.

The minority effects use Whites as the baseline for comparison. As noted earlier, these effects refer to the difference for each minority compared to Whites with regard to the chances of becoming a poor adult, regardless of class origins. For each minority group considered separately, the overall probability of becoming a poor adult may be designated as Pa, and so Pa | M refers to the percentage (i.e., probability) of becoming a poor adult for minority group M.[3] Then, Pa | W refers to the overall probability of becoming a poor adult for Whites. The net minority effect is calculated as (Pa | M–Pa | W).

## 4. Empirical Results

Table 1 shows the sample sizes for the various demographic groups, as reported by Chetty et al. (2020) in the Online Appendix, Table 5.[4] The largest group is White men, with 6,891,000. The sample sizes for Hispanic men, Black men, Asian men, and Native American men are 1,312,000, 1,348,000, 350,000, and 84,000, respectively. The sample sizes for women are similar for each racial group. The smallest sample size is 82,000 for Native American women, which is still considerable for the purpose of calculating univariate

statistics. Because of these very large sample sizes, we interpret Chetty et al.'s (2020) reported probability calculations as being substantively quite close to the actual population parameters.

**Table 1.** Sample Sizes Used by Chetty et al. (2020).

| Racial and Gender Group | Sample Size |
|---|---|
| Male | |
| White | 6,891,000 |
| Hispanic | 1,312,000 |
| Black | 1,348,000 |
| Asian | 350,000 |
| Native American | 84,000 |
| Female | |
| White | 6,599,000 |
| Hispanic | 1,303,000 |
| Black | 1,402,000 |
| Asian | 335,000 |
| Native American | 82,000 |
| Total | 19,706,000 |

Table 2 shows our calculated class effects for each race-by-gender group. The probabilities given in Column A refer to the percentage of persons born into poverty who become poor as an adult (i.e., $Pa \mid Po$). The probabilities shown in Column B refer to the percentage of persons born into affluence who become poor as an adult (i.e., $Pa \mid Ao$). These two columns of data were obtained directly from Chetty et al. (2020) in the Online Appendix, Table 5.

The class effect is shown in Column C in our Table 2. This class effect is equal to the difference between the probability of becoming a poor adult if born poor and the probability of becoming a poor adult if born affluent. That is, as noted above, the class effect equals ($Pa \mid Po$–$Pa \mid Ao$). Calculated for each racial group separately, the class effect thus indicates the disadvantage of persons born into poverty compared to persons born into affluence with regard to becoming a poor adult.

The class effects, as shown in Table 2, seem rather sizeable, with the possible exceptions of Asian men (i.e., 8.0%) and Asian women (i.e., 5.2%). The class effects for White men, Hispanic men, Black men, and Native American men are 21.3%, 15.1%, 27.0%, and 28.4%, respectively. The class effects for White men, Black men, and Native American men are thus each over twenty percent. By contrast, the difference in the class effect for Black men versus White men is 27.0–21.3%, which is equal to a fairly modest 5.7%. Native American men have the largest class effect, although it is only slightly greater than for Black men (i.e., 28.4% versus 27.0%).

Table 2 shows that, within racial groups, the class effects are smaller for women than for men. The class effects for White women, Hispanic women, Black women, and Native American women are 19.4%, 10.4%, 15.0%, and 25.1%, respectively. Native American women have the largest class effect, which is also the only one over twenty percent. However, the class effect for White women is also fairly high, being quite close to twenty percent. The difference in the class effect for Black women versus White women is 15.0–19.4%, which is equal to −4.4%, indicating that poor White women have a slightly greater class disadvantage than poor Black women.

**Table 2.** Effects of Class Origins by Race and Gender on Becoming a Poor Adult Derived from Results of Chetty et al. (2020).

| Racial and Gender Group | Probability of Poverty If Born into Poverty (%) = Pa \| Po | Probability of Poverty If Born into Affluence (%) = Pa \| Ao | Class Effect = (Pa \| Po–Pa \| Ao) |
|---|---|---|---|
| | (A) | (B) | (C) |
| Male | | | |
| White | 31.3 | 10.0 | 21.3 |
| Hispanic | 29.1 | 14.0 | 15.1 |
| Black | 48.5 | 21.5 | 27.0 |
| Asian | 19.9 | 11.9 | 8.0 |
| Native American | 49.3 | 20.9 | 28.4 |
| Female | | | |
| White | 26.7 | 7.3 | 19.4 |
| Hispanic | 20.4 | 10.0 | 10.4 |
| Black | 26.8 | 11.8 | 15.0 |
| Asian | 13.2 | 8.0 | 5.2 |
| Native American | 41.7 | 16.6 | 25.1 |

Note: Within each gender, the minority Pa | Po is statistically significant from the White Pa | Po at the 0.05 level using two-tailed tests. Within each gender, the minority Pa | Ao is statistically significant from the White Pa | Ao at any conventional level using two-tailed tests.

Table 3 shows the racial minority effects by gender. As discussed above, these effects refer to the difference for each minority compared to Whites with regard to the chances of becoming a poor adult, regardless of class origins. Since the overall probability of becoming a poor adult is designated as Pa, then Pa | M refers to the probability of becoming a poor adult for minority group M. Because Pa | W refers to the overall probability of becoming a poor adult for Whites, the minority effect may be calculated as (Pa | M–Pa | W), as was mentioned earlier.

**Table 3.** Racial Minority Effects by Gender on Becoming a Poor Adult Derived from Results of Chetty et al. (2020).

| Racial and Gender Group | Probability of Poverty for Given Minority (%) = Pa \| M | Minority Effect = (Pa \| M–Pa \| W) |
|---|---|---|
| | (A) | (B) |
| Male | | |
| White | 16.6 | — |
| Hispanic | 23.1 | 6.5 |
| Black | 39.4 | 22.8 |
| Asian | 15.8 | −0.8 |
| Native American | 37.5 | 20.9 |
| Female | | |
| White | 13.6 | — |
| Hispanic | 16.3 | 2.7 |
| Black | 21.7 | 8.1 |
| Asian | 10.4 | −3.2 |
| Native American | 31.3 | 17.7 |

Note: These statistics for Pa | M, based on Chetty et al. (2020), are reported by those authors in Tables 2 and 3 on their website: https://opportunityinsights.org/data/ (accessed on 10 October 2021). Within each gender, the minority Pa | M is statistically significant from the White Pa | M at any conventional level using two-tailed tests.

As is shown in Table 3, the minority effects are 6.5% for Hispanic men, 22.8% for Black men, and 20.9% for Native American men. The minority effects for Black men and for Native American men are thus fairly similar, while the minority effect for Hispanic men is smaller. According to the US Census official poverty measure, Hispanics have a substantially higher poverty level than Whites, but Chetty et al.'s results for the offspring

generation refer only to a particular cohort of native born, whereas low-income Hispanics are disproportionately foreign born (Orrenius et al. 2011). With regard to Asian men, their minority effect is –0.8%, which indicates that they are actually a bit advantaged over Whites with regard to becoming a poor adult, which is consistent with prior research on native-born Asian men (Takei and Sakamoto 2011).

For women, Table 3 shows that the minority effects tend to be somewhat less disadvantageous than is the case for the corresponding male group. The minority effect for Black women is 8.1%, which is notably smaller than the 22.8% for Black men. This differential is consistent with prior research reporting more negative racial effects for Black men relative to Black women (Greenman and Xie 2008; Autor et al. 2019). Although lower, the minority effect for Native American women is 17.7%, which is fairly close to Native American men. Having a minority effect of just 2.7%, Hispanic women are only slightly disadvantaged in comparison to White women. The minority effect for Asian women is −3.2%, which indicates that they are slightly more advantaged over White women than Asian men are over White men.

Table 4 shows the comparisons between the class effect and the minority effect for each of the minority-by-gender groups. The difference between these two effects is given in Column C of Table 4. For each of the groups, the class effect is larger than the minority effect. The differences for Hispanic men, Black men, Asian men, and Native American men are 8.6%, 4.2%, 8.8%, and 7.5%, respectively. Table 4 also shows that the differences for Hispanic women, Black women, Asian women, and Native American women are 7.7%, 6.9%, 8.4%, and 7.4%, respectively. For each minority group, the class effect tends to be a little lower for women than for men, but overall, these results indicate a consistent similarity across all of these groups. That is, for all of the groups, the class effect is at least several percentage points greater than the minority effect. This consistency occurs despite more variation across the groups in the class effects and the minority effects, as summarized in Columns A and C of Table 4.

**Table 4.** Comparing the Effects of Class Origins with Racial Minority Effects by Gender in Becoming a Poor Adult as Derived from Results of Chetty et al. (2020).

| Racial and Gender Group | Class Effect from Column C of Table 2 = (Pa\|Po–Pa\|Ao) | Minority Effect from Column B of Table 3 = (Pa\|M–Pa\|W) | Difference between the Class Effect and the Minority Effect = |
|---|---|---|---|
| | (A) | (B) | (C) |
| Male | | | |
| White | 21.3 | — | — |
| Hispanic | 15.1 | 6.5 | 8.6 |
| Black | 27.0 | 22.8 | 4.2 |
| Asian | 8.0 | −0.8 | 8.8 |
| Native American | 28.4 | 20.9 | 7.5 |
| Female | | | |
| White | 19.4 | — | — |
| Hispanic | 10.4 | 2.7 | 7.7 |
| Black | 15.0 | 8.1 | 6.9 |
| Asian | 5.2 | −3.2 | 8.4 |
| Native American | 25.1 | 17.7 | 7.4 |

Note: For each minority-race-by-gender group, the class effect is statistically significant from the minority effect at any conventional level using two-tailed tests.

## 5. Discussion and Conclusions

Sakamoto and Kim (2010) analyzed the differentials on productivity versus annual earnings. Their results indicate substantial underpayment for workers in the lowest quintile of the distribution of annual earnings. That is, relative to their contributions to economic productivity, workers in the lowest quintile were underpaid by about 53%. By contrast, workers in the highest quintile were overpaid by about 162% relative to their contributions

to economic productivity. Sakamoto and Kim (2010) interpret these findings as evidence for rising economic exploitation that is associated with workers in the lower versus the upper ends of the earnings distribution. The latter contributes directly to inequality in the distribution of household income, in which the upper quintile currently obtains 50.5% of the total income while the lowest quintile receives 3.6% (Semega et al. 2020, p. 45). These and other related results of class inequality (e.g., Leicht 2016; Sakamoto and Kim 2014) are completely ignored; however, there are many studies in critical race theory that emphasize race as the dominant factor relating to poverty (e.g., Alexander 2012; Bonilla-Silva 2004, 2015; Bonilla-Silva and Baiocchi 2001; Bonilla-Silva and Dietrich 2011; Christian et al. 2019, 2021; Feagin 2013, 2020; Ray et al. 2017).

In the foregoing, we have extended the study of poverty to intergenerational mobility, broken down by race-by-gender categories. Children from the lowest quintile are much more likely to become poor adults than are children from the upper quintile. As shown in Table 2, this class effect is substantial (i.e., double-digit in terms of percentage points) for all of the races by gender groups, with the slight exception of Asians, who are known for an exceptionally high level of upward mobility (Chetty et al. 2020; Sakamoto and Wang 2021). Tables 2 and 3 further indicate that the class effects and the minority effects tend to be slightly less disadvantageous for females as compared to males for each racial category. With regard to gender differentials, the most notable group is Blacks, among whom the men are significantly more likely than White men to become poor adults, and that differential is, furthermore, significantly greater than the higher chances of Black women becoming poor compared to White women.[5]

Nonetheless, for all of the race-by-gender groups, the class effect is greater than the minority effect, as shown in Table 4. This conclusion also applies to Black men, among whom the class effect is larger than among White men, Hispanic men, Asian men, and any of the female groups, including Native American women. Overall, our findings indicate that the effects of class inequality on intergenerational mobility are large as well as consistently greater than the minority effects. Poverty and racial inequality are often discussed synonymously in terms of "white supremacy", as if the class inequalities within racial groups are too trivial to mention (e.g., Bonilla-Silva 2004, p. 941). However, most income inequality is within racial groups and not between them (Sakamoto and Wang 2015), and our analysis reveals that class effects on intergenerational mobility are substantial for all race-by-gender groups, except perhaps Asians.

These conclusions are consistent with prior arguments that state that an increased focus on the expansion of civil rights laws alone will not significantly reduce poverty in the United States (Sakamoto and Wang 2015; Wilson 1980). Studies of critical race theory have become immensely popular in recent years (Douchet 2021), influencing not only sociology, but also criminal justice, popular media and entertainment, workplace training, legal and political discourse, educational reforms at all levels, and popular political movements, such as "Black Lives Matter" (Butcher and Gonzalez 2020). Our results that find large class effects for all racial groups clarify that reducing racial discrimination, per se, is clearly not sufficient to eliminate the substantial intergenerational transmission of poverty. While our research objective has not been to critique critical race theory, our analysis nonetheless demonstrates the continuing significance of large class effects in determining economic disadvantages for all racial groups in the contemporary United States.

**Author Contributions:** Conceptualization, A.S.; methodology, M.E.J.; software, L.H.; validation, A.S.; formal analysis, L.H; investigation, M.E.J.; writing—original draft preparation, A.S.; writing—review, A.S.; supervision, A.S.; project administration, A.S. All authors have read and agreed to the published version of the manuscript.

**Funding:** This research received no external funding.

**Institutional Review Board Statement:** Not applicable.

**Informed Consent Statement:** Not applicable.

**Data Availability Statement:** Not applicable.

**Conflicts of Interest:** The authors declare no conflict of interest.

## Notes

[1] According to Google Scholar, as retrieved on 28 May 2022.

[2] Our use of the term "poverty" is also loose because we do not adjust for household size, as is often done (Smeeding 2016). Our research objective is based on the results of Chetty et al. (2020), who do not use any equivalence scaling.

[3] Chetty et al. (2020) do not report Pa for any of the groups. These figures are instead available in Tables 2 and 3, located on the webpage associated with their publication (https://opportunityinsights.org/data/) (accessed on 10 October 2021).

[4] The Online Appendix for Chetty et al. (2020) does not contain any page numbers. Chetty et al. (2020) report these sample sizes to the nearest 1000.

[5] For related discussion and analysis of this issue, see Autor et al. (2019).

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
