# Peer review of "Comparing the Effects of Class Origins versus Race in the Intergenerational Transmission of Poverty"

_socsci, doi:10.3390/socsci11060257_

Round 1
Reviewer 1 Report
This manuscript “Comparing the Effects of Class Origins versus Race in the Intergenerational Transmission of Poverty” compares, as the title suggests, class and race effects on the probability of becoming a poor adult in the United States. The article is effectively written and edited well, and it engages with an important question.
The article’s main weakness is that it does not present any new theoretical insights, and it does not collect or analyze original data, either. It looks as though three of the paper’s four results tables are taken pretty much directly from the Chetty et al. (2020) study. The author(s) are clear and transparent about this.
Parts of the manuscript conflate economic inequality and poverty (for example, in the first paragraph of the “discussion and conclusions” section), so it would be helpful to differentiate these more in the manuscript’s discussions. Finally, the manuscript is limited in exploring practical implications of its findings and, although the writing is clear overall, the manuscript could use some additional light editing.
My recommendation would be to revise the manuscript and add in an original data analysis alongside the Chetty et al. findings. The original analysis would not need to be nearly as extensive as the Chetty et al. dataset, but its insights would make a much more compelling case for publication.
Author Response
- Reviewer 1 identified an important limitation of the previous version of the paper which is the lack of clarity regarding the theoretical significance of the empirical analysis. To rectify this problem, the introduction has been extensively re-written and expanded. Many more relevant references have been cited. As is now explained, our analysis is pertinent to recent research in critical race theory which views class inequalities among minority groups as being unproblematic because their poverty is viewed as being entirely due to racial discrimination. As is now discussed in the new introduction as well as in the expanded discussion section, our analysis is the first paper to calculate class effects for minority groups in regard to the chances of becoming a poor adult.
- Placed in the above theoretical context, the term "class" is still used to differentiate from the issue of racial discrimination. In the revised version, we more clearly note that our analysis only is considering one aspect of class inequality, namely, the differential between being born into a poor family as opposed to being born into an affluent family. This point is explicitly reiterated at the beginning of the Data and Methods section.
- We believe that the theoretical significance of our analysis is now more clearly substantiated in the text. In any event, adding additional new analysis using Chetty's data is not legally possible because we do not have legal access to confidential U.S. Census and IRS records. The uniqueness and magnitude of Chetty's study is not explicitly discussed in the second paragraph of the Data and Methods section. Even if we had legal access, the resources required to assemble and analyze hundreds of millions of data records is beyond the scope of any single researcher because a great deal of research funding is required to support such a project.
- The policy implications of our findings are now discussed in the last paragraph of the Discussion and Conclusion section.
Reviewer 2 Report
I consider that the article is worth publishing. In general, I see few issues that could be clarified or corrected to further improve the quality of the article.
- I consider that it would be good that the formulas occupied to calculate the statistics in the tables were presented in a brief subsection.
- In the introduction the authors textually mention that
“In that regard, we define the class effect as the difference between the probability that a person who was born into the lowest income quintile becomes a poor adult and the probability that a person who was born into the highest income quintile becomes a poor adult.”
It would be good that the authors explained more about this definition, particularly in statistical terms.
- It may also be good that Chetty et al.’s statistical method be discussed deeper.
Author Response
- We agree with Reviewer's 2 excellent suggestion to organize and explain the statistical formulas so that they are more clearly comprehensible in one section. The final two paragraphs of the Data and Methods section now do that.
- The class effect is more thoroughly explained in the third and fourth paragraphs in the Introduction of the revised version. The class effect is explained again substantively in the Data and Methods section. In addition, a new paragraph has been added to the Data and Methods section which explains the notation used in the statistical formula for calculating the class effect.
- Some further discussion of Chetty et al.'s (2020) methods has been added to the Data and Methods section. This discussion has also been moved up earlier on in that section in order to make it more prominent. However, we did not add too much new discussion of Chetty's methods because too much detail is not necessary to understand our analysis and too much detail could become a distraction to readers.
Round 2
Reviewer 1 Report
The author(s) have done a good job of expanding the paper's theoretical discussion and connecting the research to existing scholarship. More discussion of implications at the end is also helpful. The paper is very close to being ready for publication, but it just needs another round of edits to catch minor errors.